# A Combination of Heavy Metals and Intracellular Pathway Modulators Induces Alzheimer Disease-like Pathologies in Organotypic Brain Slices

**DOI:** 10.3390/biom14020165

**Published:** 2024-01-30

**Authors:** Dhwani S. Korde, Christian Humpel

**Affiliations:** Laboratory of Psychiatry and Experimental Alzheimer’s Research, Medical University of Innsbruck, 6020 Innsbruck, Austria; dhwani.korde@i-med.ac.at

**Keywords:** Alzheimer, amyloid-beta, tau, organotypic brain slices, heavy metals, intracellular modulators, slice model, tauopathy, spreading

## Abstract

Alzheimer’s disease (AD) is a progressive neurodegenerative disorder that is characterized by amyloid-beta (Aβ) plaques and tau neurofibrillary tangles (NFT). Modelling aspects of AD is challenging due to its complex multifactorial etiology and pathology. The present study aims to establish a cost-effective and rapid method to model the two primary pathologies in organotypic brain slices. Coronal hippocampal brain slices (150 µm) were generated from postnatal (day 8–10) C57BL6 wild-type mice and cultured for 9 weeks. Collagen hydrogels containing either an empty load or a mixture of human Aβ42 and P301S aggregated tau were applied to the slices. The media was further supplemented with various intracellular pathway modulators or heavy metals to augment the appearance of Aβ plaques and tau NFTs, as assessed by immunohistochemistry. Immunoreactivity for Aβ and tau was significantly increased in the ventral areas in slices with a mixture of human Aβ42 and P301S aggregated tau compared to slices with empty hydrogels. Aβ plaque- and tau NFT-like pathologies could be induced independently in slices. Heavy metals (aluminum, lead, cadmium) potently augmented Aβ plaque-like pathology, which developed intracellularly prior to cell death. Intracellular pathway modulators (scopolamine, wortmannin, MHY1485) significantly boosted tau NFT-like pathologies. A combination of nanomolar concentrations of scopolamine, wortmannin, MHY1485, lead, and cadmium in the media strongly increased Aβ plaque- and tau NFT-like immunoreactivity in ventral areas compared to the slices with non-supplemented media. The results highlight that we could harness the potential of the collagen hydrogel-based spreading of human Aβ42 and P301S aggregated tau, along with pharmacological manipulation, to produce pathologies relevant to AD. The results offer a novel ex vivo organotypic slice model to investigate AD pathologies with potential applications for screening drugs or therapies in the future.

## 1. Introduction

### 1.1. Alzheimer’s Disease

Alzheimer’s disease (AD) is a debilitating, progressive neurodegenerative disorder that accounts for the majority (60–70%) of all dementia cases. It is characterized by the extracellular deposition of plaques composed of amyloid-β (Aβ) and intracellular accumulation of neurofibrillary tangles (NFTs) consisting of hyperphosphorylated tau protein [1]. Aggregated tau protein manifests as neuropil threads and dystrophic neurites. These are degenerated axons containing tau in the vicinity of Aβ plaques [1].

Given the multifactorial nature of AD, research in the disease etiology and pathology has proven challenging. Animal models have been the gold standard for AD research with rodent models being the primary choice to model pathological features of familial AD [2,3]. For instance, the transgenic (TG) mouse model overexpressing the amyloid precursor protein (APP) harboring the Swedish–Dutch–Iowa mutations is a well-characterized model that recapitulates extensive Aβ plaques [4]. TG mice expressing non-mutant human tau isoforms exhibit tau pathology, including hyperphophorylated tau accumulation and cognitive deficits [5,6]. Evidence from in vitro studies utilizing two- and three-dimensional models has furthered our understanding of the AD pathology [7,8,9,10].

### 1.2. Study of AD-Like Pathologies in Organotypic Brain Slices

Organotypic brain slices (100–400 µm) from mouse or rat brains can be cultured for months using the membrane interface-slice culture method by Stoppini and colleagues. Thus, brain slice cultures are an attractive methodological approach in neurodegeneration research [11,12,13]. This approach preserves the three-dimensional brain cytoarchitecture and provides access to all cell types in the brain, unlike cell lines, primary cell cultures or iPSC cultures. Cells and synapses develop in a manner comparable to the in vivo environment [14]. This system allows direct and repeated manipulation, including the addition of exogenous compounds to the medium or application of collagen hydrogels to achieve a slow, localized protein or peptide release [15,16,17,18,19]. The absence of a blood–brain barrier permits direct investigation of the effects of toxic or therapeutic compounds [20].

### 1.3. Culture of Postnatal Brain Slices

Most organotypic brain slice cultures use postnatal day 8–12 mouse or rat brains due to advanced cytoarchitecture, ease of handling, and improved neuronal cell survival rates [11,12,21]. Slice cultures from wild-type (WT) animals are extensively employed to study AD pathology, starting from early studies introducing exogenous Aβ in culture media to induce AD-like pathologies [22]. Previous reports demonstrate the presence of “plaque-like deposits” in rat brain slices cultured for a period of 12 weeks in low pH conditions with apolipoprotein E4 [23]. Some models use TG mouse models of AD, such as TgCRND8 mice, to generate slices that recapitulate AD pathology [24,25,26,27]. Tau pathology has been modelled using postnatal organotypic brain slices from various TG models (3xTg-AD, JNPL3, TauRDΔK and hTau), showing increased levels of phosphorylated tau and thioflavin-S positive tau inclusions from 1 to 4 weeks in culture and redistribution of tau to the somatodendritic compartments [28,29,30,31,32]. However, a slice culture model from WT postnatal mice encompassing both AD pathologies of Aβ plaques and tau-containing NFTs has not been reported.

### 1.4. Culture of Adult Brain Slices

Age is a critical risk factor for most neurodegenerative diseases [33]. Organotypic brain slices from adult animals offer exploration of disease-relevant neuropathological processes, including access to aged neurons. Adult brain slices are appealing for testing therapeutics. Thin sections (120 µm) from adult AD mice overexpressing APP with the Swedish–Dutch–Iowa mutations (APP_SDI) cultured for two weeks recapitulated in vivo pathology, exhibiting thioflavin S positive-Aβ plaques and gliosis [34]. Slices from adult tau TG (P301S) animals were cultured for two weeks using thinner slices, slightly hypothermic temperatures and 5% CO_2_ incubation conditions [35]. Serum-free media and a gliogenesis inhibitor sustained adult hippocampal organotypic slices for at least three weeks in culture [36]. Despite methodological advances, culturing adult organotypic brain slices remains challenging.

### 1.5. Use of Exogenous Human Spreading on Mouse Brain Slices

Aβ and tau pathologies spread stereotypically throughout the brain with different stages aligning with disease severity [37]. The prion hypothesis postulates systemic aggregation and propagation of Aβ and tau pathologies via interconnected pathways, akin to prion protein transmission [38,39,40,41,42]. Slice cultures have been used to study this pathology transmission [16,17]. Our prior research on postnatal WT animals showed increased spreading of P301S aggregated tau (aggTau) to the ventral areas of slices via neuroanatomically connected pathways [15]. Sagittal slices from a TG mouse model of Parkinson’s disease overexpressing α-synuclein exhibited enhanced spreading of α-synuclein pre-formed fibrils compared to WT mice [19]. The potent spreading activity of human Aβ42 (hAβ42) was reported in a slice co-culture paradigm with microglial activation helping clear hAβ42 aggregates [18].

Building on our prior work, we aim to establish an organotypic brain slice model simulating both Aβ-containing plaques and tau-containing NFT pathology. As mouse models do not produce plaques per se, we plan to apply exogenous hAβ42 and P301S aggTau to generate AD-like pathologies. Using our established collagen hydrogel loading system, we induce the spreading of the exogenously applied peptides/proteins, followed by applying various pharmacological post-treatments to enhance AD-like pathologies. Additionally, we intend to extend these experiments to slices from adult WT and TG mice. Our data reveal that combining intracellular pathway modulators and heavy metals in the media with the induced spreading of hAβ42 and P301S aggTau leads to increased Aβ plaque- and tau NFT-like immunoreactivity in ventral regions of WT postnatal slices.

## 2. Materials and Methods

### 2.1. Animals

This study used wild-type (WT, C57BL/6N) and transgenic APP_SweDI (TG APP_SDI, expressing the amyloid precursor protein harboring the Swedish K670N/M671L, Dutch E693Q and Iowa D694N mutations) mice. They were housed in the animal facility at the Centre for Chemistry and Biomedicine with open access to food and water under 12/12 h light-dark cycles. Adults (1 male, 1 female) were housed per cage and the pups were kept in the same cage until postnatal day 8–10. The animals used for the organotypic brain slices were randomized between the groups, irrespective of sex. The Austrian Ministry of Science and Research (BMWF-66.011/0120-II/3b/2013 and 66.011/0055-WF/V/3b/2017) approved all animal experiments. The ethical principles of the 3R rules were followed in our experiments and slice preparation is defined as organ removal as opposed to animal experiments. All animal-related lab work complied with the Austrian and international guidelines on animal welfare and experimentation. The fixed coronal brain cryosections of transgenic mice expressing human tau (hTau) were commercially obtained from QPS Austria GmbH (Grambach, Austria). These mice express human tau derived from a human PAC, H1 haplotype, also known as 8c mice, while murine tau is knocked out by a targeted disruption of exon 1 [13]. They have a hybrid background of C57/blk6, DBA, Swiss Webster, C57BL/6 and 129/SvJae.

### 2.2. Organotypic Brain Slice Cultures

Organotypic brain slices were prepared as reported in detail in previous studies [25,29]. Briefly, postnatal C57BL/6N pups (day 8–10) or adult WT or TG APP_SDI (6–8 months) were rapidly decapitated. The brains were extracted under sterile conditions inside a laminar flow hood. The cerebellum was manually dissected with a sharp razor blade and the brains were glued (Glue Loctite 401) onto the chuck of a water-cooled vibratome (Leica Biosystems, Nussloch, Germany, VT1000A). Coronal slices (110/150 μm) were cut at the hippocampal level using a commercial razor blade in a sterile preparation medium (16 mg/mL MEM/HEPES (Gibco, Thermo Fisher Scientific, Vienna, Austria, 11012-044), 0.43 mg/mL NaHCO_3_ (Merck-Millipore, Darmstadt, Germany, 144-55-8), pH 7.2). Two slices were carefully placed directly onto 0.4 μm membrane inserts (Merck-Millipore, Darmstadt, Germany, PICM03050) in a sterile 6-well plate prepared on the day (Greiner-Merck, Darmstadt, Germany, 657160) (see Figure 1A in Results). The excess sterile preparation medium was carefully removed from the insert.

Each well contained 1 mL of sterile culture medium (16 mg/mL MEM/HEPES (Gibco, Thermo Fisher Scientific, Vienna, Austria, 11012-044), 0.43 mg/mL NaHCO_3_ (Merck-Millipore, Darmstadt, Germany, 144-55-8), 6.25 mg/mL glucose (Merck-Millipore, Darmstadt, Germany, 1083371000), 25% Hank’s solution (Gibco, Thermo Fisher Scientific, Vienna, Austria, 24020-091), 10% heat-inactivated horse serum (Gibco, Thermo Fisher Scientific, Vienna, Austria, 16050-122), 1% glutamine 200 mM stock solution (Merck-Millipore, Darmstadt, Germany, 1002890100), and 1% of antibiotic-antimycotic (Gibco, Thermo Fisher Scientific, Vienna, Austria, 15240062, pH 7.2)). The slices were incubated for a maximum of 9 weeks at 37 °C and 5% CO_2_ with frequent checks to ensure continuity of culture conditions. The medium was changed weekly until the slices were fixed in 4% paraformaldehyde (PFA) for 3 h and stored in PBS/0.1% Na-Azide at 4 °C until further use.

Media containing heavy metals and intracellular pathway modulators was prepared with the desired final concentration and sterile filtered before use. The following heavy metals were utilized at a final concentration of 100 nM- aluminum chloride (Sigma Aldrich-Merck, Darmstadt, Germany, 206911), lead acetate (Sigma Aldrich-Merck, Darmstadt, Germany, 316512), iron sulphate (Sigma Aldrich-Merck, Darmstadt, Germany, 307718), and cadmium chloride (Sigma Aldrich-Merck, Darmstadt, Germany, 202908). The following intracellular pathway modulators were used- okadaic acid (100 nM; Santa Cruz, sc-3513), wortmannin (10 nM; Sigma Aldrich-Merck, Darmstadt, Germany, W1628), scopolamine (50 nM; Sigma Aldrich-Merck, Darmstadt, Germany, Y0000483), and MHY1485 (50 nM; Sigma Aldrich-Merck, Darmstadt, Germany, SML0810), ApoE4 (10 ng/mL; Sigma Aldrich-Merck, Darmstadt, Germany, A3234).

### 2.3. Proteins/Peptides

We used the following proteins/peptides in this study: recombinant human tau (mutated P301S) protein aggregate (active) (Abcam, Cambridge, UK, ab246003), human Aβ42 (hAβ42, Innovagen AP-BA-42-1, Lund, Sweden) and recombinant human tau (full-length tau, R&D Systems, Minneapolis, MN, USA, SP-495).

Recombinant human full-length tau was aggregated by incubating 600 ng of the protein in 10 μL of EDTA-free Protease Inhibitor Cocktail (PIC, Sigma Aldrich-Merck, Darmstadt, Germany, P-8340) and 10 μL of 55 μM heparin solution with 74 μL of PBS in 100 μL of aggregation solution. The solution was incubated for 72 h at 37 °C. Human Aβ42 was aggregated by dissolving it in 1 mg/mL Tris-HCl (pH 9.0), diluted 1 + 1 with PBS + 0.05% sodiumdodecylsulfate, giving a concentration of 100 µM, and incubated overnight at 4 °C. It was then diluted again 1:10 with PBS and incubated for two weeks at 4 °C, yielding a final concentration of 50 µg/mL (10 µM). The efficiency of the full-length tau and hAβ42 aggregation was checked using Western blot analysis (see Appendix A).

### 2.4. Collagen Hydrogels and Loading of hAβ42 and P301S aggTau

Collagen hydrogels were prepared as described previously [25,29]. As a cross-linker, a solution of 4arm-poly (ethylene glycol) succinimidyl succinate (4arm-PEG; Sigma Aldrich-Merck, Darmstadt, Germany, JKA7006) was used, which was prepared by dissolving 2.5 mg of 4S-StarPEG in 400 µL of 10 mM phosphate-buffered saline (PBS, pH 7.2) for a stock concentration of 6.25 mg/mL. From this solution, 0.16 mg of 4S-StarPEG was mixed with 0.4 mg of 3 mg/mL type I bovine collagen solution (Sigma Aldrich-Merck, Darmstadt, Germany, 804592) with 20 µL of 100 mM PBS. A load of 20 µL of either 1 mg/mL hAβ42 or 0.1 mg/mL P301S aggTau was added in a total final volume of 200 μL. The same amount of 10 mM PBS was added in place of the hAβ42 and P301S aggTau peptides/proteins to produce empty collagen hydrogels as negative controls. Samples were kept on ice throughout to prevent premature polymerization of the hydrogels. The pH of the collagen hydrogel solution was set to 7.2 and 2 μL of the solution was pipetted on UV-sterilized Teflon-tape-coated glass slides for gelation. Subsequently, the collagen hydrogels were incubated for 15 min at 37 °C followed by immediate application onto organotypic brain slices (see Figure 1B,C in Results).

### 2.5. Immunohistochemistry

Immunostainings were performed as detailed in prior studies [25,29]. Fixed brain slices were washed 3× in PBS and incubated in PBS-0.1% Triton (T-PBS) for 30 min at 20 °C on a shaker. Slices were then incubated with PBS-20% methanol-1% H_2_O_2_ to quench endogenous hydrogen peroxidase binding for 20 min at 20 °C when using biotinylated secondary antibodies. Following washing the slices 3× with PBS, they were blocked with T-PBS-0.2% bovine serum albumin (BSA, Serva, Heidelberg, Germany, 11930.03)-20% horse serum for 30 min at 20 °C. An extra blocking step for the primary antibodies raised in mice was performed by incubating the slices with PBS-mouse on mouse (M.O.M., Vector Laboratories, Newark, CA, USA, MKB-2213) at a concentration of 1 drop per 2.5 mL for 1 h at 20 °C. Afterwards, slices were incubated with primary antibodies diluted in T-PBS-0.2% BSA for 2 days at 4 °C.

The following antibodies were utilized to detect tau and Aβ immunoreactivity: Tau5 (1:250, Invitrogen, Thermo Fisher Scientific, Vienna, Austria, AHB0042), Aβ antibody clone 6E10 (1:1000, BioLegend, Vienna, Austria, 803015) and PHF-Tau monoclonal antibody/AT8 (1:250, Thermo Fisher Scientific, Vienna, Austria, MN1020). These antibodies are characterized in additional experiments (see Appendix A). Subsequently, the slices were washed with PBS and incubated with the appropriate conjugated biotinylated (diluted by 1:200) or fluorescent secondary antibodies (diluted by 1:400) in T-PBS- 0.2% BSA for 1 h at 20 °C while shaking. When using a biotinylated secondary antibody, the slices were rinsed 3× with PBS and incubated with avidin–biotin complex solution (Elite ABC-HRP Kit, Vector Laboratories, Newark, CA, USA, PK-6100) for 1 h at 20 °C. Finally, the slices were washed with 50 mM Tris-buffered saline (TBS) and then incubated in 0.5 mg/mL 3,3′-diaminobenzidine (DAB, Sigma Aldrich-Merck, Darmstadt, Germany, D7304)-TBS-0.003% H_2_O_2_ at 20 °C in the dark until a signal was detected. Upon the appearance of a DAB signal, the reaction was stopped by adding TBS. The slices were washed 3× in PBS and mounted with Mowiol onto gelatin-coated glass slides.

When using a fluorescent secondary antibody, the slices were washed 3× with PBS prior to and following incubation with either Alexa-488 (green fluorescent, Invitrogen, Thermo Fisher Scientific, Vienna, Austria, A11029) or Alexa-546 (red fluorescent, Invitrogen, Thermo Fisher Scientific, Vienna, Austria, A11030/A11040) antibodies. All slices were counterstained with the blue fluorescent nuclear dye DAPI (1:10,000). After a final washing step with PBS, slices were mounted with Mowiol onto glass slides. Immunostainings were visualized with a light microscope (Olympus BX61, Vienna, Austria). Images were captured and analyzed with OpenLab software (Version 5.5.0, Improvision Ltd., Perkin Elmer, Vienna, Austria).

Fluorescence microscopy was conducted using a confocal laser scanning microscope with an AiryScan detector (Zeiss LSM980 AiryScan, Germany) with a 1.2 NA glycerol objective. The emission of AlexaFluor-488 was detected from 488 to 527 nm and Thiazine Red dye was detected from 510 to 580 nm. Subsequently, images and z-stacks were captured and deconvoluted by Huygens Professional software (Version 23.10.0, Scientific Volume Imaging, Hilversum, Netherlands) and reconstructed in three dimensions with Imaris software (Version 8.2, Oxford Instruments, Abingdon, UK).

### 2.6. Processing of Human Post-Mortem Tissue Sections

To probe for the appearance of Aβ plaques and tau NFTs, immunohistochemistry was performed on formalin-fixed paraffin-embedded (FFPE) tissue from the temporal lobe of a human patient diagnosed post-mortem with AD (Biochain, Newark, CA, USA, T1236078Alz). Tissue slides with a thickness of 5–10 µm were incubated for 1 h at 60 °C and deparaffinized in acetic acid n-butyl ester (Carl Roth, Karlsruhe, Germany, P036.1) for 20–30 min until the paraffin was visibly removed. Subsequently, the tissue sections were rinsed in 99% ethanol and incubated in 0.45% H_2_O_2_ in 99% ethanol for 30 min at room temperature (RT) to block endogenous peroxidase activity. Then, the tissues were sequentially hydrated in 96%, 90% and 70% ethanol for 5 min each and washed 2× with distilled water. Heat-mediated antigen retrieval was performed by incubating the sections in 10 mM sodium citrate buffer with 0.05% Tween-20 overnight at 37 °C. The sections were cooled to RT the next day and then the immunohistochemistry protocol was followed as outlined above.

### 2.7. Propidium Iodide Live Staining

To measure cell death, slices were incubated with sterile-filtered slice media containing the nuclear dye propidium iodide at a concentration of 2 µg/mL for 30 min at 37 °C. Following the incubation, the slices were washed twice with PBS and fixed with 4% PFA for 3 h and kept in the dark until further immunofluorescence experiments.

### 2.8. Western Blot

Immunostainings were carried out as described in prior studies [25,29]. Organotypic brain slices were carefully scraped off the cell culture inserts and collected in Eppendorf tubes. The slices were mixed with 80–120 µL of EDTA-free PIC (Sigma Aldrich-Merck, Darmstadt, Germany, P-8340) and sonicated using an ultrasonic device (Hielscher Utrasonic Processor, Teltow, Germany). Then, the sonicated mixture was centrifuged at 14,000× *g* for 10 min at 4 °C and the supernatant was collected. The total amount of protein in the supernatant sample was determined by performing a Bradford assay with Coomassie brilliant blue G-250 dye (Bio-Rad, Vienna, Austria, #1610406).

Samples were loaded onto 10% Bis-Tris polyacrylamide gels (Invitrogen, Thermo Fisher Scientific, Vienna, Austria, NP0301BOX) following denaturation (10 min at 70 °C with 2 μL of sample-reducing agent (Invitrogen, Thermo Fisher Scientific, Vienna, Austria, NP000). Thereafter, electrophoresis was carried out at 200 V for 35 min. The gel was assembled together with a PDVF membrane (Merck, Darmstadt, Germany, ISEQ00010) with layers of blotting paper and the samples were electrotransferred for 20 min at 25 V in a semi-dry transfer cell (Thermo Fisher Scientific, Vienna, Austria). The blotting steps utilized reagents from the WesternBreeze Chemiluminescent immunodetection system (Invitrogen, Thermo Fisher Scientific, Vienna, Austria). Blots were blocked with a blocking buffer for 30 min and then incubated overnight at 4 °C. The following primary antibodies were used: Neurofilament (1:5000, Proteintech, Rosemont, IL, USA, 60331-1-Ig) and glial fibrillary acidic protein (GFAP) (1:2000, Merck-Millipore, Darmstadt, Germany, AB5541). Next, the blots were washed and incubated with alkaline-phosphatase-conjugated secondary antibodies (anti-mouse for neurofilament, anti-chicken for GFAP) for 30 min at RT. Following brief washing of the blots, they were treated with CDP-Star chemiluminescent substrate solution (Roche, Basel, Switzerland) for 15 min and visualized with a cooled CCD camera (SearchLight, Thermo Fisher Scientific, Vienna, Austria).

### 2.9. Data Analysis and Statistics

Images were acquired at 10× or 20× magnification from the light microscope using the same brightness and automatic exposure time settings from OpenLab image acquisition software (Version 5.5.0). Images from the 10× and 20× magnification have a field size of 1089 × 817 µm and 523 × 392 µm, respectively. They were analyzed on a blinded basis using ImageJ (1.53 k, National Institutes of Health, Bethesda, MD, USA). The background was subtracted using a rolling ball radius of 30–50 pixels, depending on the image analyzed.

The multi-point tool in ImageJ was utilized to count the number of cells in a given image. To count the fiber densities in the images, a predefined grid (3 × 3 cm) was overlaid and the number of times a strongly stained fiber would cross the grid was recorded. For measuring the number of Aβ plaques, the background was subtracted from images. Images were thresholded using either the Triangle or Default thresholding algorithms in ImageJ and then, subject to despeckling and adding watershed demarcations. The particles/Aβ plaques in the image were counted in the size range of 100–8000 pixels and circularity of 0.2–1.0. For optical density analysis, the mean value of the selected fields was recorded from the ventral part of the slices. Two images from the left and right hemispheres were captured and analyzed per slice and the values from both hemispheres were averaged per slice. The final optical density was calculated by subtracting the slice background of each image from the initial values. Statistical analysis was performed by one-way ANOVA with a Fisher’s LSD post hoc test, where *p* < 0.05 represented significance. A student’s *t*-test with equal variance was used when comparing two groups. Data values are presented as mean ± standard error of mean (SEM), unless stated otherwise. Values in parentheses in bar graphs denote the number of analyzed animals in each experimental group.

## 3. Results

### 3.1. Culturing of Organotypic Brain Slices and Collagen Hydrogel Application

Coronal sections were produced and collected on cell culture inserts (150 µm thickness) (Figure 1A). All slices were qualitatively assessed with regard to tissue thinning, adherence to the membrane of the cell culture inserts, and damaged slices were excluded from experiments. Within one week of culture, healthy slices adhere well to the cell culture insert and appear glossy (Figure 1A). Subsequently, collagen hydrogels loaded with hAβ42 peptides or P301S aggTau protein were prepared and applied to the left and right cortical locations. After 9 weeks in culture conditions, the spread of these peptides/proteins was evaluated in the ventral areas as indicated by the red dotted outlines (Figure 1B). Collagen hydrogels could be immunohistochemically detected on top of brain slices. For instance, hydrogels loaded with hAβ42 were detectable as a concentrated circle at the site of application using an anti-Aβ antibody upon fixation after 1 week in culture (Figure 1C). Aβ and tau were characterized through immunoblotting (see Appendix A).

**Figure 1 biomolecules-14-00165-f001:**
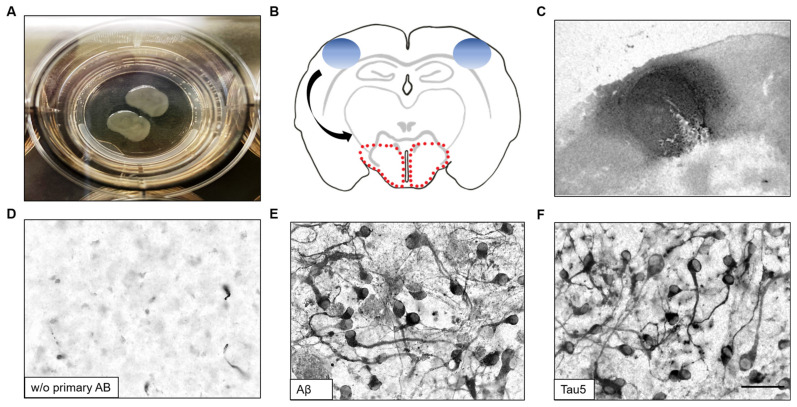
Spreading of human amyloid-beta (hAβ42) and P301S aggregated tau (aggTau) to ventral areas. Slices are generated at the hippocampal level from postnatal mice (day 8–10). (**A**) A representative image of coronal organotypic brain slices on cell culture inserts. (**B**) A schematic image of a brain slice with the location of the applied collagen hydrogels on the left and right cortical areas (blue ovals). Spreading of hAβ42 and P301S aggTau was quantified in the ventral most regions of the slices as demarcated by the red dotted outlines. (**C**) Collagen hydrogels can be visualized on the slice after 1 week in culture using the Aβ antibody clone 6E10 to detect Aβ immunoreactivity. (**D**) The absence of primary antibodies in the immunohistochemistry of slices showed a background staining. (**E**) A representative image of Aβ-like immunoreactivity in the ventral area of interest from slices incubated with collagen hydrogels with hAβ42 for 8 weeks in culture. (**F**) Spreading of P301S aggTau was visualized using the Tau5 antibody in slices cultured with collagen hydrogels with P301S aggTau for 8 weeks. Scale bar in F = 18 mm in (**A**); 465 µm in (**C**); 50 µm in (**D**–**F**).

### 3.2. Spreading of hAβ42 and P301S aggTau to Ventral Areas

To examine the spread of hAβ42 and P301S aggTau peptides/proteins, coronal hippocampal slices were cultured with collagen hydrogels loaded with either hAβ42 or P301S aggTau for a total of 9 weeks. Upon fixation and immunohistochemistry, Aβ- and tau-like immunoreactivity was analyzed in the ventral areas from the left and right hemispheres. A representative image of a slice shows no immunoreactivity without a primary antibody as a negative control (Figure 1D). On the contrary, slices incubated with hAβ42- and P301S aggTau-loaded hydrogels show clear Aβ-like (Figure 1E) and tau-like immunoreactivities (Figure 1F) with demarcated neuronal structures and staining is observed in the cell bodies and along axons.

### 3.3. Aβ Plaques and Tau NFTs in Postnatal WT and TG APP_SDI Slices

We sought to investigate whether we could harness the exogenous delivery of hAβ42 and P301S aggTau through collagen hydrogels to induce Aβ plaques and tau NFTs in slices from WT and TG APP_SDI postnatal mice. Collagen hydrogels containing either an empty load (PBS), hAβ42, P301S aggTau or a combination of both were applied on slices after 1 week and further cultured for 8 weeks. The presence of Aβ-like and tau-like immunoreactivity was quantified in ventral regions as detected by the Aβ antibody clone 6E10, Tau5 and AT8 antibodies.

In WT slices incubated with hAβ42-loaded collagen hydrogels, a significantly increased number of Aβ+ cells and Aβ+ fiber density were measured compared to the slices incubated with empty collagen hydrogels (*p* < 0.05 for both parameters, Figure 2A,B). Similarly, in slices with P301S aggTau-loaded collagen hydrogels, Aβ+ cell numbers and fiber density were significantly raised versus the negative control slices (*p* < 0.001, *p* < 0.01, respectively, Figure 2A,B). Additionally, a slightly elevated number of Aβ+ cells were recorded in the slices with both hAβ42 and P301S aggTau as opposed to P301S aggTau alone (*p* < 0.05, Figure 2A). The presence of P301S aggTau also significantly augmented the number of Aβ+ cells and Aβ+ fiber density in the combination hydrogel compared to only hAβ42, suggesting an influence of the aggregated tau on the spread and accumulation of hAβ42 in the ventral regions (*p* < 0.001, *p* < 0.01, respectively, Figure 2A,B). On the other hand, no significant differences in the number of Aβ+ cells or fibers could be seen within groups in TG APP_SDI slices (Figure 2A,B). No mature plaques were visible in the postnatal slices either from the WT or TG APP_SDI mice.

An increased number of Tau5+ cells and fiber density were observed in the WT slices incubated with the P301S aggTau-loaded hydrogels compared to the control group (*p* < 0.001, *p* < 0.05, respectively, Figure 2C,D). Slices with a mixed load of hAβ42 and P301S aggTau display an elevated number of Tau5+ cells and fiber density to slices with only hAβ42 (*p* < 0.001, *p* < 0.05, respectively, Figure 2C,D). The presence of hAβ42 does not influence the Tau5+ immunoreactivity in WT slices. Similar patterns were seen in the TG APP_SDI mice wherein Tau5+ cells and fiber density were significantly increased in the slices with P301S aggTau-loaded collagen hydrogels compared to control slices (*p* < 0.05 for both parameters, Figure 2C,D). Moreover, there was a significant increase in Tau5+ cells in the hAβ42 + P301S aggTau versus the hAβ42 group, denoting that in this specific experimental setup, the hAβ42 did not affect the cellular accumulation of tau aggregates (*p* < 0.05, Figure 2C). However, Tau5+ fiber density was significantly increased in the slices with hAβ42 than the empty collagen hydrogels (*p* < 0.05, Figure 2D). No tau-NFTs were detected in either WT or TG APP_SDI slices using the AT8 antibody.

### 3.4. Aβ Plaques and Tau NFTs in Post-Mortem Human and TG Mice Slices

To examine the appearance of Aβ plaques and tau NFTs as positive controls, we performed immunohistochemistry with post-mortem sections from the temporal lobe of a human AD patient and TG APP_SDI and TG mice expressing the human tau protein. The absence of primary antibody served as negative controls for the human post-mortem and TG mice slice immunostainings with no apparent staining indicative of plaques or tangles (Figure 3A,E). Plenty of senile plaques positive for Aβ were visible in human post-mortem tissue sections with a characteristic dense plaque core and the outer corona (Figure 3B). Moreover, NFTs could be identified in human post-mortem sections with compact staining for phosphorylated tau inside the neuronal cytoplasm with unstained nuclei. The NFTs were dispersed frequently throughout the human PM slice (Figure 3C). Aβ plaques and NFTs could be successfully visualized by fluorescent immunostaining using Thiazine red dye for plaques and AT8 antibody for detecting NFTs. The classical “flame-shaped” NFT morphology was visible, which colocalised with the DAPI nuclear counterstain along with an Aβ plaque in the vicinity (Figure 3D). Abundant Aβ plaques could be detected in slices from adult TG APP_SDI mice (7–9 months old, Figure 3F). In the cryosections from TG mice expressing human tau, phosphorylated tau containing NFTs were visualized in the ventral regions using both DAB and fluorescence immunohistochemistry (Figure 3G,H). These immunostainings established the appearance of Aβ plaques and tau NFTs using our antibodies of choice for this study (Aβ antibody clone 6E10, AT8).

### 3.5. Pharmacological Manipulation of Slices to Generate Aβ Plaques and Tau NFTs

Given that we did not observe Aβ plaques and tau NFTs in postnatal slices from WT and TG APP_SDI mice, we hypothesized that treatment of WT slices with pharmacological agents combined with collagen hydrogel with hAβ42 and P301S aggTau could potentiate AD-like pathology. Slices from WT postnatal mice were prepared and collagen hydrogel containing hAβ42 and P301S aggTau were applied after 1 week. The slices were cultured for 4 weeks and, subsequently, low concentrations of either intracellular pathway modulators (Okadaic acid, wortmannin, scopolamine, MHY1485, ApoE4) or heavy metals (Aluminum, lead, cadmium, iron) were separately supplemented in slice media for 4 weeks. The Aβ- and AT8-like immunoreactivities were evaluated by quantifying optical densities in the ventral areas of the slices.

From the intracellular pathway modulators, scopolamine and MHY1485 media significantly increased Aβ-like immunoreactivity compared to non-supplemented slice media (minus group, *p* < 0.05, *p* < 0.01, respectively, Figure 4A). The heavy metals groups, aluminum, lead, and cadmium, significantly elevated Aβ-like immunoreactivity as opposed to the minus group (*p* < 0.05, *p* < 0.01. *p* < 0.001, respectively, Figure 4A). Slices incubated with wortmannin, scopolamine and MHY1485 displayed a significantly increased AT8-like immunoreactivity compared to slices incubated with no pharmacological agent in the media (*p* < 0.001 for all treatments, Figure 4B). The addition of okadaic acid to the media proved to be detrimental for the slices as the AT8-like immunoreactivity significantly reduced compared to the control slices (*p* < 0.01, Figure 4B). The addition of heavy metals, specifically aluminum, lead and iron, slightly augmented AT8-like immunoreactivity (*p* < 0.05 for all groups, Figure 4B).

To investigate whether an additive effect on Aβ- and AT8-like immunoreactivity could be observed upon using a mixture of the pharmacological agents, WT slices were incubated with different combinations of intracellular pathway modulators and/or heavy metals (Figure 4C,D). Upon quantification of the optical densities in the ventral regions of the slices, combining MHY1485 and cadmium significantly increased Aβ-like immunoreactivity versus control slices, however, not in comparison to only cadmium-supplemented media. A mixture of lead and cadmium significantly raised Aβ-like immunoreactivity (*p* < 0.001 for both groups, Figure 4C). Combining scopolamine, wortmannin and MHY1485 resulted in a significant increase in AT8-like immunoreactivity versus control slices (*p* < 0.001, Figure 4D).

Next, we explored whether combining all five pharmacological agents could boost the appearance of Aβ- and AT8-like immunoreactivity. This group consisted of scopolamine, wortmannin, MHY1485, lead and cadmium (combination treatment). Slices incubated with this group exhibited significantly increased Aβ- and AT8-like immunoreactivity, pointing to an additive effect of combining various pharmacological agents to generate AD-like pathologies in WT slices (*p* < 0.001, Figure 4C,D).

### 3.6. Plaque-Like and NFT-Like Features upon Treatment with Pharmacological Agents

Representative images from DAB immunohistochemistry showed a background with no apparent staining when postnatal slices from the mixed pharmacological agent group (scopolamine, wortmannin, MHY1485, lead, cadmium) were processed without a primary antibody (Figure 5A). On the contrary, clear Aβ-like immunoreactivity could be seen in neuronal cells, specifically inside the cell bodies (Figure 5B). An intense immunostaining for AT8-like immunoreactivity was observed in the cytoplasm of neuronal cells with densely stained axons intertwining within the field of magnification (Figure 5C). Representative images of treatment with other pharmacological agents display varying levels of Aβ- and AT8-like immunoreactivity (see Appendix A). High-resolution confocal microscopy was performed and z-stacks obtained from the ventral parts of slices and then reconstructed in 3-D. A representative image of this reconstruction showed Aβ plaque-like immunoreactivity (Figure 5D). Tau NFT-like immunoreactivity was detected in neuronal cells in the somatic compartment and along axons (Figure 5E). Both pathologies were detectable when the slices were stained with thiazine red dye for Aβ plaques and AT8 antibody for tau NFT-like pathology along with DAPI as a nuclear counterstain (Figure 5F).

### 3.7. Aβ Plaques Develop Intracellularly Prior to Cell Death

Based on the 3-D reconstruction of z-stacks with Aβ plaque-like staining with thiazine red dye, we hypothesized that Aβ first accumulates inside neurons prior to its release. To investigate this further, slices were incubated with a mixture of hAβ42 and P301S aggTau and a combination of pharmacological agents (scopolamine, wortmannin, MHY1485, lead, cadmium). Prior to fixation, the slices were incubated with propidium iodide to mark for cell death. Immunohistochemistry with the Aβ antibody indicated healthy looking cells with Aβ+ immunoreactivity and a clumped structure resembling a plaque in the vicinity (Figure 6A). PI staining showed a few dead cells in the field of magnification (Figure 6B). Viable nuclei were detected with the DAPI nuclear counterstain (Figure 6C). The merged image revealed that the Aβ+ immunoreactivity in the plaque-like structure colocalizes with the dead cells as marked with PI (Figure 6D). A positive control used for PI staining was included by adding hydrogen peroxide to slices prior to fixation (see Appendix A).

### 3.8. Translation to Adult WT and TG APP_SDI Slices

Using slices from adult animals permits an exploration in a more disease-relevant environment, considering the influence of age as a risk factor for AD. To this end, we examined if the results obtained from postnatal slices could be translated to adult slices. Adult WT and TG APP_SDI animals (6–7 months old) were utilized to generate thinner hippocampal coronal slices (110 µm) to increase the viability of the tissue for long-term culture. The zoomed-in image displays a representative AT8+ staining in the ventral areas (Figure 7A).

We determined the overall viability of adult slices by Western blotting. Acute adult slices from WT animals were collected and immediately frozen at −80 °C. Adult WT slices were cultured for 9 weeks. The blot exhibits a strong signal for neurofilament and GFAP (neuronal and astroglial marker, respectively) in the fresh slices compared to the culture slices (Figure 7B). A faint signal can be seen in the cultured slices for neurofilament and GFAP, indicating an extensive neuronal and glial loss and a loss of viability after the long culturing period.

Similar to the postnatal slices, collagen hydrogels containing either an empty load, hAβ42 or P301S aggTau were applied on the adult slices from WT and TG APP_SDI animals and the spreading of these peptides/proteins was quantified in ventral areas. Only in adult slices from TG APP_SDI, AT8+ cells were significantly augmented in slices incubated with either hAβ42 or P301S aggTau versus the empty hydrogel group (*p* < 0.05, *p* < 0.01, respectively, Figure 7C).

We examined whether the combination treatment of lead, cadmium, scopolamine, wortmannin and MHY1485 would result in the appearance of Aβ plaque- and tau NFT-like pathologies in adult slices. Hippocampal adult slices were cultured with hAβ42 and P301S aggTau containing hydrogels for 4 weeks and treated with media supplemented with the combination of pharmacological agents for another 4 weeks. Upon quantification of Aβ and AT8-like immunoreactivity in ventral areas using optical density as a proxy measure, no significant difference was observed between slices incubated with normal culture media versus the combination treatment media (Figure 7D).

## 4. Discussion

In this study, our goal was to create an ex vivo AD model with Aβ plaques and tau NFTs. We employed collagen hydrogels to deliver hAβ42 and P301S aggTau to organotypic brain slices to induce spreading. Intracellular pathway modulators and heavy metals were used to enhance Aβ plaques and tau NFTs appearance. Results show that heavy metals preferentially boosted Aβ plaque-like pathology, developing intracellularly before neuronal death. Intracellular pathway modulators strongly increased tau NFT-like pathology. A combination of scopolamine, wortmannin, MHY1485, lead, and cadmium significantly increased both Aβ plaque- and tau NFT-like immunoreactivity in ventral areas. Both these pathologies could develop independently.

### 4.1. Spreading of hAβ42 and P301S aggTau in Postnatal Slices

In WT slices with hAβ42-loaded collagen hydrogels, a significantly increased number of Aβ+ cells were observed in the ventral region compared to empty collagen hydrogels. Combining hAβ42 and P301S aggTau slightly elevated Aβ+ cell numbers compared to P301S aggTau alone. This suggests potent spreading activity of hAβ42, detectable away from the application site in the ventral parts of the slice, aligning with previous reports on hAβ42 spread to connected target slices [18].

Combining P301S aggTau with hAβ42 increased Aβ+ immunoreactivity in ventral areas in terms of Aβ+ cells and fiber density compared to hAβ42-loaded hydrogels alone, suggesting an interplay between mutant tau and hAβ42 spreading. This aligns with the notion that tau is essential for cytotoxicity in primary neurons and Aβ alone (small oligomers or fibrillary form) is insufficient for the induction of neurodegeneration [43,44,45,46]. However, the tau-Aβ interplay in AD is complex, with bidirectional influence on each other’s aggregation and toxicity [47]. No plaques were detectable in postnatal slices from WT or APP_SDI mice, possibly due to reported plaque occurrence starting from 3 months in this model [4]. Yet, hippocampal slice cultures from a 3xTg-AD model show Aβ42 and hyperphosphorylated oligomeric tau accumulation by 28 days in vitro [25,29]. Similar accumulation levels were found in vivo starting at 12 months, indicating that slice culture potentiates Aβ plaques and tau NFTs formation based on the accelerated accumulation.

Tau5+ cells and fiber density increased in WT and TG APP_SDI slices with P301S aggTau-loaded hydrogels compared to control slices. This aligns with the hypothesis that exogenous tau aggregates boost tau+ immunoreactivity, consistent with previous findings of increased tau+ immunoreactivity in the ventral parts of slices from WT postnatal mice [15]. The presence of hAβ42 does not impact tau accumulation in WT slices, as evidenced by the lack of a significant difference in immunoreactivity between slices incubated with hAβ42 and P301S aggTau versus P301S aggTau alone. No tau-containing NFTs were detected in WT or APP_SDI slices using the AT8 antibody. This could be because the slices were generated from postnatal animals (D8–10) and cultured for a maximum of 9 weeks, which may be a premature time point for NFT formation. Additionally, no tangles are reported in the APP_SDI model [4].

### 4.2. Aβ Plaque- and Tau Tangle-Like Pathologies Develop Independently

Data from postnatal WT and TG APP_SDI showed an independent appearance of Aβ- and tau-like pathologies in slices incubated with either peptide/protein loads in collagen hydrogels. This challenges the amyloid cascade hypothesis, which postulates that Aβ deposition triggers downstream tau pathology leading to synaptic loss, neurodegeneration and AD dementia [48]. The initial sites of Aβ and tau accumulation and spreading in AD patients are spatially and temporally distinct, with tau pathology initiating in the transentorhinal cortex and locus coeruleus whilst Aβ initially accumulates in association cortices and neocortical regions [49,50,51,52]. Tau NFTs can be found without Aβ plaques in primary age-related tauopathy (PART) [53]. This underscores that our results are in line with such observations.

We observed increased Tau5+ cells and fiber density in postnatal WT and TG APP_SDI slices with exogenously added P301S aggTau. This suggests that aggregated tau pathology can spread and accumulate independently of Aβ, in line with studies indicating tau tangles precede Aβ plaque formation in post-mortem autopsies [54].

### 4.3. Heavy Metals Augment Aβ Plaque-Like Pathology

As Aβ plaques and tau NFTs were not observed in postnatal slices from WT or TG APP_SDI mice with exogenous hAβ42 and P301S aggTau alone, we explored other triggers for inducing AD pathology in organotypic brain slices. Considering the increasing focus on lifestyle and other environmental factors in AD development, chronic exposure to heavy metals is of particular interest given the widespread exposure of the population [55,56,57]. Postnatal WT mouse slices were incubated with hAβ42 and P301S aggTau, along with nanomolar concentrations of non-essential metals (aluminum, lead, cadmium) and one essential metal (iron) in the slice media. Treatment with aluminum, lead, or cadmium significantly increased Aβ+ immunoreactivity in ventral areas compared to control slices, highlighting the influence of these heavy metals on the accumulation of Aβ pathology.

Although aluminum is not an essential metal for living organisms, it exerts a myriad of biologically relevant actions in the mammalian brain, including neurotransmitter synthesis and protein phosphorylation or dephosphorylation. It specifically influences the Aβ deposition and aggregation along with inhibiting peptide degradation [58,59,60,61,62]. Lead exposure reportedly increases APP and Aβ levels in animal models, including primates [63,64,65]. Cadmium’s link to neurotoxicity in AD is well established with reports demonstrating Aβ aggregation, plaque deposition and interactions with Aβ in transgenic rodent models [66,67]. Our results align with these studies, evident in the increased Aβ+ immunoreactivity in ventral regions of the slices.

### 4.4. Intracellular Pathway Modulators Boosts tau NFT-Like Pathology

We hypothesized that agents modulating intracellular pathways implicated in Aβ and tau pathogenesis would induce plaque and tangle pathology in slices. Okadaic acid is involved in tau hyperphosphorylation via the selective inhibition of serine/threonine phosphatases 1 and 2A and contributes to learning and memory deficits in rodent models [68,69,70,71,72]. A previous study from our lab utilized okadaic acid (100 nM) to hyperphosphorylate tau in adult organotypic brain slices for 2 weeks [73]. However, using the same concentration in our experiments for an extended culture period of 4 weeks proved too toxic, resulting in unhealthy slices with a lack of viable cells.

Wortmannin selectively inhibits phosphatidylinositol 3-kinase (PI3K) and activates glycogen synthase kinase-3 β (GSK3-β), resulting in tau hyperphosphorylation [74,75,76]. In our experiments, nanomolar wortmannin concentration significantly increased tau NFT-like immunoreactivity, consistent with a previous report in rat brain hippocampal slices, showing increased tau hyperphosphorylation with micromolar wortmannin concentration [77]. We opted for nanomolar wortmannin concentration to preserve slice viability over the 4-week culture period. In vivo injections of wortmannin in rats also induced tau hyperphosphorylation, emphasizing its ability to directly modulate intracellular pathways through PI3K and GSK-3β [78].

Scopolamine increased both Aβ+ and tau+ immunoreactivities in slices, suggesting a role in neuropathological changes. Widely used to induce memory or cognitive deficits in rodent models for dementia-related studies [79], scopolamine-driven neurodegeneration in rats is reported, including increased Aβ protein, APP mRNA levels, phosphorylated tau, and GSK-3β levels [80]. It is implicated in oxidative stress, mitochondrial dysfunction, and neuroinflammatory processes [81]. Since we assessed only Aβ plaque- and tau tangle-like pathologies, scopolamine treatment may induce other cellular modifications relevant to AD pathophysiology.

This study implicates MHY1485 in the accumulation of Aβ plaque- and tau NFT-like pathologies in slices, marking the first such observation to our knowledge. MHY1485 is an activator of the mammalian target of rapamycin (mTOR). The hyperactivation of the mTOR signaling pathway and its downstream players is increasingly implicated in AD pathogenesis [82,83]. The results support the hypothesis that activating the mTOR pathway can potentiate Aβ and tau pathologies in postnatal slices, suggesting the involvement of mTOR in Aβ and tau-induced AD neurodegeneration.

Surprisingly, no significant differences in Aβ plaque- and tau NFT-like pathologies were observed in slices treated with human ApoE4, despite its crucial genetic role as a consistent risk factor for AD development [84]. This may be attributed to the concentration used, and further explorations with varying concentrations are warranted for a more comprehensive understanding of AD-relevant pathologies.

### 4.5. Combined Model of AD Neuropathologies

We investigated the additive effect of combining heavy metals and intracellular pathway modulators on postnatal slices treated with collagen hydrogels containing hAβ42 and P301S aggTau. Combining lead and cadmium with wortmannin, scopolamine, and MHY1485 resulted in a significant increase in Aβ+ and AT8+ immunoreactivities compared to non-supplemented media, suggesting an additive effect. Confocal microscopy revealed phosphorylated tau-containing neurons near Aβ+ plaque-like structures.

Other slice models of AD usually exhibit either Aβ or tau pathology. For instance, recombinant adeno-associated viruses (rAAV) were utilized to transduce tau variants in brain slices and recapitulated mature neurofibrillary tau inclusions [28]. However, such a model enables the study of tau pathology alone without Aβ pathology. It requires further training in handling rAAVs and time to set up the correct plasmids for transfection. On the other hand, our model displays both AD-relevant pathologies in ventral areas, highlighting that these neuropathologies can be examined in different brain areas apart from the hippocampus, which remains highly investigated in AD. This combination model is cost-effective and easily set up utilizing WT postnatal mouse tissue. It not only contributes to the 3R’s principle of reducing the number of animals in research but also surpasses the need to age mice to study neurodegenerative processes. The addition of heavy metals and intracellular pathway modulators is easy and different concentrations can be combined. The use of nanomolar concentrations ensures minimal impact on cellular viability. Lastly, culturing slices from postnatal mice for over 2 months allows exploration of the spreading of amyloid and tau pathology, as demonstrated in previous studies from our lab [15,18,19].

### 4.6. Aβ Plaques Develop Intracellularly before Cell Death

Colocalization with propidium iodide revealed intracellular Aβ pathology preceding cell death. Propidium iodide, indicating compromised membrane integrity, is a reliable marker for cell death in organotypic brain slices [18,85,86].

Intracellular Aβ accumulation has been reported in endosomes and multivesicular bodies in transgenic rodent models and human brains before extracellular deposition [87,88,89,90,91,92,93]. Our results show clumped plaque-like structures with comprised nuclei near healthy cells, positively stained for Aβ. This shows that Aβ accumulates intracellularly in organotypic brain slices, leading to neuronal lysis and toxic Aβ release to form plaques. Intracellular Aβ42 was detected in pyramidal neurons from post-mortem human AD brain tissue where the lysis of a single neuron resulted in Aβ42-positive plaque formation [91]. Another study showed fibrillar Aβ42 in perinuclear compartments as a precursor to neuritic plaque formation after the death of intracellular Aβ-containing neurons in a 3xTg-AD mouse model and human AD post-mortem tissue [94]. Thus, intracellular Aβ may drive early AD pathogenesis and our postnatal WT mouse slice model allows studying these processes.

### 4.7. Translation to Adult Slices

We investigated if our earlier findings on hAβ42 and P301S aggTau spread and pathology induction through pharmacological manipulation could be replicated in adult slices. Adult slices (110 µm) from WT and TG APP_SDI mice used in a previous study from our lab showed that thinner adult slices permit a longer culturing period [34]. To our knowledge, this is the first study to culture adult slices for 2 months, mimicking the culturing period of postnatal slices and allowing sufficient time for hAβ42 and P301S aggTau spread from collagen hydrogels. Despite the relatively low thickness of the slices, adult slices did not flatten over time. Western blotting analysis revealed reduced neurofilament and GFAP expression in cultured adult slices, indicating decreased cellular viability. This result agrees with previous studies reporting substantial cell loss (>90%) within 14 days for adult slices [35,95,96].

Significantly increased AT8+ cells were observed in ventral areas of adult TG APP_SDI slices incubated with exogenous hAβ42 or P301S aggTau, suggesting the potential of inducing tau pathology in adult slices. This contrasts with previous reports of no tau pathology in this mouse model [4]. The outcome resonates with the prior observations using postnatal slices treated with P301S aggTau-loaded collagen hydrogels [15]. In adult slices, hAβ42-loaded collagen hydrogels led to a significant increase in AT8+ immunoreactivity, indicating differential responses in aged/mature neurons to exogenous hAβ42.

However, when investigating the combined treatment of heavy metals and intracellular pathway modulators, no significant differences in Aβ+ or AT8+ immunoreactivities were observed between slices incubated with empty media and those with media supplemented with nanomolar concentrations of scopolamine, wortmannin, MHY1485, lead, and cadmium. The lack of significance may be attributed to widespread neuronal loss or overall decline in cell viability during the 2-month culture period of adult slices. Despite using identical culture conditions as postnatal slices, it is likely that viable neurons are crucial for the accumulation, spread, and development of Aβ and tau pathology. Further advancements in culturing adult slices for extended periods will enhance the study of AD-relevant processes in slices with sustained cellular viability.

### 4.8. Translation to Humans and Outlook for Therapeutic Strategies

Through the application of exogenous hAβ42 and P301S aggTau, along with the treatment of heavy metals and intracellular pathway modulators, we developed an ex vivo model incorporating the two primary pathologies in AD. However, due to the multifactorial nature of AD etiology, our model has limitations in fully replicating all human neuropathologies, restricting it to research questions at this stage and it does not have a direct translation to humans.

Although our model is a proof-of-principle exploration, it proves valuable for preclinical drug screening. Specifically, multiple slices from a single animal can be generated, allowing simultaneous testing of various compounds with biochemical or histological analyses. For preclinical testing, Aβ and tau aggregation inhibitors can be assessed in this amenable system. Slice culture is a powerful tool to screen for potential AD drugs, such as BTA-EG4 that reduced tau phosphorylation [97], rhodamine-based tau aggregation inhibitor bb14 [32], and curcumin that counteracted the deleterious effects of exogenous Aβ42 addition [98]. Thus, our slice model offers a practical approach to test and evaluate drugs or therapeutic substances for AD.

### 4.9. Limitations of the Study

This study has limitations. (1) We tested a single concentration of heavy metals and intracellular pathway modulators to induce Aβ plaque- and tau NFT-like pathologies, potentially overlooking the impact of varying concentrations on Aβ and tau pathologies. (2) Our ex vivo model focused on Aβ and tau pathologies in slices, neglecting assessment of other AD-relevant markers like neuroinflammation or reactive glial cells, which increasingly have a prominent role in AD pathogenesis [99]. A full characterization of the model is essential to ascertain if other pathologies are also recapitulated. (3) The spread of exogenous hAβ42 and P301S aggTau in the TG APP_SDI model was analyzed, which overexpresses APP. It would be intriguing whether another TG model that incorporates both amyloid and tau pathology, such as 3xTg-AD model (APP Swedish, MAPT P301L, and PSEN1 M146V), could provide a more relevant environment [100]. (4) The majority of AD cases (>90%) are sporadic and late-onset as clinical symptoms manifest over 60 years of age. However, the preclinical stage typically lasts for 20–30 years. The familial early-onset AD cases with a genetic basis represent 1–2% of cases [101]. Considering the remarkable amount of time for neuropathologies to show up as clinical symptoms, the 9-week culture period in our experiments limits comparisons to in vivo situations. (5) Organotypic brain slices inherently lack possibilities for systemic or behavioral assessments, restricting their ability to substitute the in vivo environment.

## 5. Conclusions

This study demonstrates the spread and accumulation of hAβ42 and P301S aggTau in the ventral areas of postnatal organotypic brain slices. Both pathologies could develop independently, challenging the classical amyloid-beta cascade hypothesis. Heavy metals (aluminum, lead, cadmium) in the culture media augment Aβ plaque-like pathology that develops intracellularly before cell death. Intracellular pathway modulators (scopolamine, wortmannin, MHY1485) in the culture media preferentially boosted the accumulation of tau NFT-like pathologies. Combining both heavy metals and intracellular pathway modulators significantly induces Aβ and tau pathologies. Hence, combining the spreading of exogenous protein aggregates and pharmacological manipulation provides a valuable ex vivo model to study the finer details of AD pathology, screen drugs or therapeutic compounds.

## Figures and Tables

**Figure 2 biomolecules-14-00165-f002:**
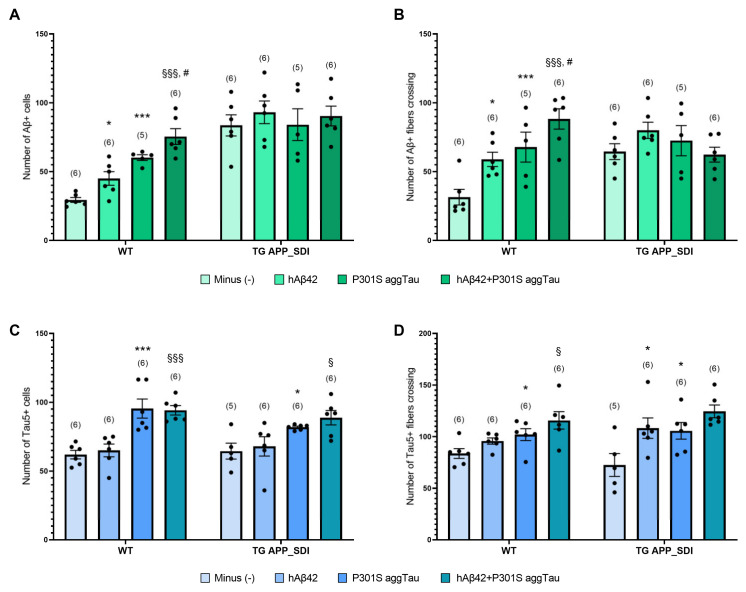
Spreading of human amyloid-beta (hAβ42) and P301S aggregated tau (aggTau) in postnatal wild-type and transgenic organotypic brain slices. Brain slices were prepared from postnatal day 8–10 wild-type (WT) C57BL6 mice or transgenic (TG) amyloid precursor protein _Swedish–Dutch–Iowa (APP_SDI) mice. Collagen hydrogels with hAβ42, P301S aggTau, or a mix of both was loaded after 1 week in culture. Collagen hydrogels with an empty load were included as negative controls. Slices were cultured for 8 weeks, fixed and analyzed by immunohistochemistry for Aβ using the antibody clone 6E10 or for tau using the antibodies Tau5 or AT8. Images were taken at the 20× magnification with a field size of 523 × 392 µm, which were quantified using ImageJ (Version 5.5.0). (**A**) Quantification of the number of Aβ+ cells in the ventral areas of WT and TG APP_SDI slices. (**B**) Quantification of the Aβ+ fiber density. (**C**) Quantification of the number of Tau5+ cells in the ventral areas of WT and TG APP_SDI slices. (**D**) Quantification of the Tau5+ fiber density. The corresponding fiber densities were calculated by counting the number of times fibers crossed a predefined grid on the same image used to analyze the number of Aβ+ or Tau5+ cells. Note that no mature Aβ plaques or tau neurofibrillary tangles were observed in these postnatal slices. Values are given as mean ± SEM for each group and the values in parentheses represent the number of analyzed animals. The dots represent individual raw data values. Statistical analyses were performed using a one-way ANOVA with a Fisher’s LSD post-hoc test. * *p* < 0.05, *** *p* < 0.001 signifies comparisons against the respective (−) groups. § *p* < 0.05, §§§ *p* < 0.001 signifies comparisons between hAβ42 + P301S aggTau and hAβ42 groups. # *p* < 0.05 signifies comparisons between hAβ42+ P301S aggTau and P301S aggTau groups.

**Figure 3 biomolecules-14-00165-f003:**
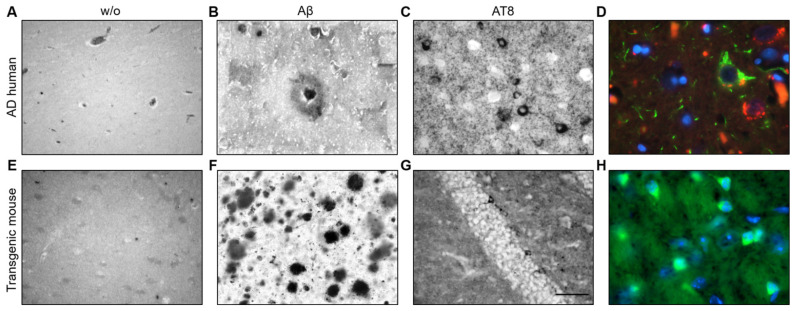
Characterization of amyloid-beta (Aβ) plaques and tau neurofibrillary tangles (NFT) in post-mortem human and transgenic mouse slices. Tissue sections from the temporal lobe of a human Alzheimer’s disease (AD) patient (**A**–**D**) were probed for the appearance of Aβ plaques and tau NFT. Following the deparaffinization and antigen retrieval protocols, the slices were subject to immunohistochemistry. (**A**) The lack of a primary antibody (w/o) served as a negative control with background staining. (**B**) An Aβ plaque was detected by using the Aβ antibody clone 6E10 with a dense plaque core and the surrounding corona. (**C**) Tau containing NFTs were visualized in human tissue sections with a densely stained cytoplasm and unstained nuclei. (**D**) A representative image displayed the pyramid-shaped tau NFT (green; AlexaFluor-488), Aβ plaques (red; Thiazine Red dye) and nuclei (blue; DAPI). (**E**) Slices from transgenic (TG) amyloid precursor protein _Swedish–Dutch–Iowa (APP_SDI) mice (**F**–**H**) exhibited no staining without a primary antibody (w/o). (**F**) Slices from TG APP_SDI mice displayed numerous Aβ plaque structures. (**G**) Slices from a TG mouse model expressing the human tau protein were commercially obtained and probed for tau NFTs, which could be detected with intracellular staining and unstained nuclei. (**H**) A representative image from the human tau-expressing TG mice showed strong staining in neuronal cells detected by AT8 antibody (green; AlexaFluor-488) with nuclear staining (blue; DAPI). Scale bar in G = 50 µm in (**A**–**C**); 27 µm in (**D**); 100 µm in (**E**–**G**); 27 µm in (**H**).

**Figure 4 biomolecules-14-00165-f004:**
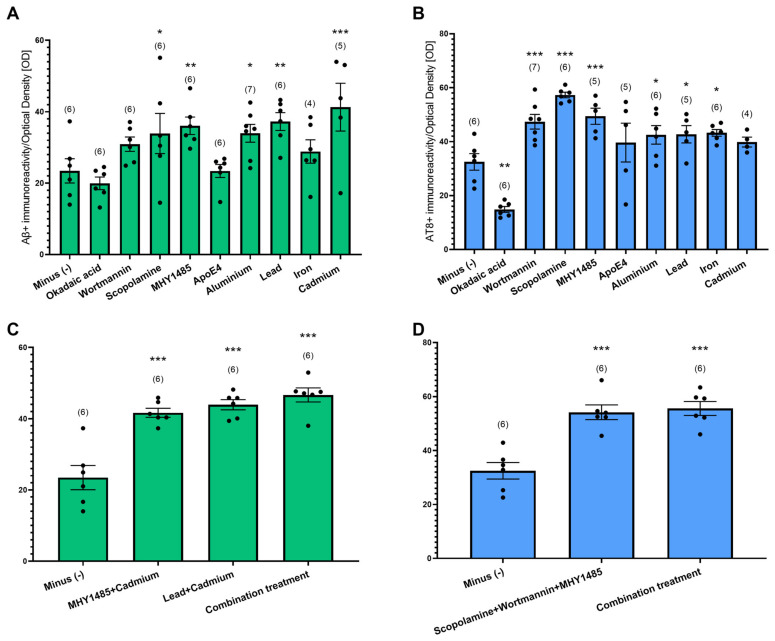
Pharmacological manipulation of postnatal wild-type organotypic brain slices using intracellular pathway modulators and heavy metals. Organotypic slices at the hippocampal level from postnatal wild-type (WT) mice were prepared. After 1 week in culture, collagen hydrogels loaded with a combination of human amyloid-beta 42 (hAβ42) and P301S aggregated tau (aggTau) were applied. After 4 weeks of culture, the pharmacological treatments were supplemented in the media at a final concentration of 100 nM for the heavy metals, 50 nM for scopolamine and MHY1485, 10 nM for wortmannin, 100 nM for okadaic acid, and 10 ng/mL for ApoE. The slices were cultured for a total of 9 weeks. Slices were fixed and immunostained with the Aβ antibody clone 6E10 and AT8 antibodies. Images were taken at 10× magnification with a field size of 1089 × 817 µm with the same exposure and light settings and then quantified using ImageJ. (**A**) Quantification of the Aβ+ immunoreactivity in the ventral slice regions. (**B**) Quantification of the AT8+-like immunoreactivity was evaluated in ventral areas of the slices. (**C**) Quantification of the Aβ+ immunoreactivity in the slices with combination treatments. (**D**) Quantification of the AT8+ immunoreactivity in the slices with combination treatments. Raw optical density measurements were inverted by correcting for slice background for each image such that 0 represents white and 255 represents black. Values are generated from an average of two slices per animal and values in parentheses indicate the number of analyzed animals. Values are reported as mean ± SEM of optical density value. The dots represent individual raw data values. Statistical analyses were performed using a one-way ANOVA with a Fisher’s LSD post-hoc test, where *p* values < 0.05 represent significance versus the Minus (−) group (* *p* < 0.05, ** *p* < 0.01, *** *p* < 0.001).

**Figure 5 biomolecules-14-00165-f005:**
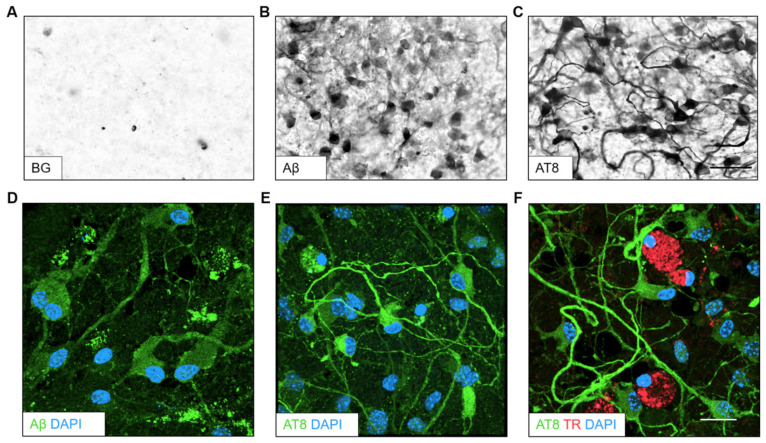
A combination of intracellular pathway modulators and heavy metals induce amyloid-beta (Aβ) and tau neurofibrillary tangle (NFT)-like immunoreactivities in postnatal wild-type mouse slices. Slices from postnatal wild-type (WT) mice (day 8–10) were generated and collagen hydrogels containing both human amyloid-beta 42 (hAβ42) and P301S aggregated tau (aggTau) were applied after 1 week. After 4 weeks in culture, a combination of scopolamine, wortmannin, MHY1485, lead and cadmium was prepared in sterile slice media (50, 10, 50, 100, 100 nM final concentration, respectively) and added to the slices and they were further cultured for 4 weeks. Slices were fixed and processed by immunohistochemistry for Aβ+- and AT8+-like immunoreactivity. (**A**) No primary antibody in the process yields a background staining. (**B**) Aβ+ like immunoreactivity was observed in the ventral areas of the slices. (**C**) Strongly stained AT8+-like immunoreactivity was recorded in the slices treated with the exogenous hAβ42 and P301S aggTau and a mixture of pharmacological agents. (**D**) A representative image of the Aβ plaque-like immunoreactivity from the 3-D reconstruction of confocal microscopy z-stacks stained for Aβ42 (green) and the nuclei (blue). (**E**) A representative image of the tau NFT-like immunoreactivity (green) and the nuclei (blue). (**F**) A representative image highlighting Aβ plaque-like pathology (red) by using the thiazine red (TR) dye. Tau NFT-like pathology (green) was visualized using the AT8 antibody whilst DAPI stains were used for nuclei (blue). Scale bar in C = 50 µm in (**A**–**C**); scale bar in f = 10 µm in (**D**–**F**).

**Figure 6 biomolecules-14-00165-f006:**
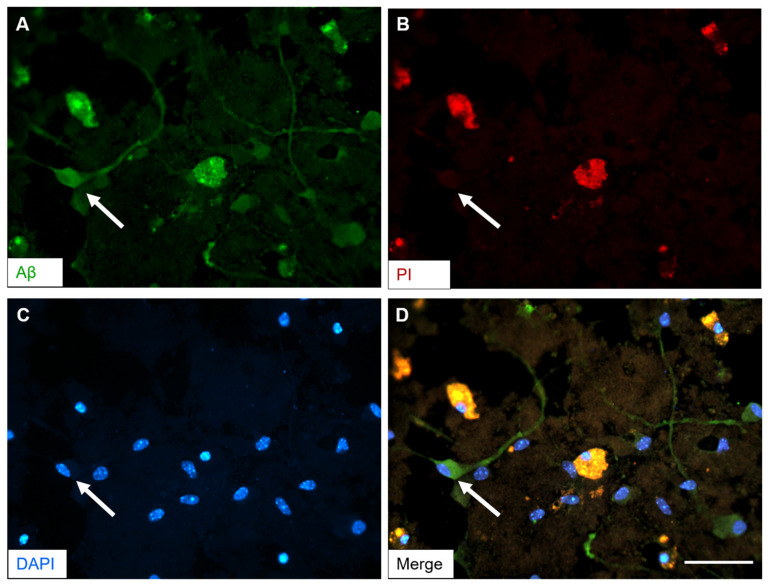
Amyloid-beta (Aβ) plaques develop intracellularly prior to cell death. Organotypic brain slices were produced from postnatal (day 8–10) wild-type mice and collagen hydrogels with human Aβ42 and P301S aggregated tau were applied. After 4 weeks of a culturing period, the media was supplemented with scopolamine, wortmannin, MHY1485, lead and cadmium (50, 10, 50, 100, 100 nM final concentration, respectively). The slices were cultured for another 4 weeks. Prior to fixation, the slices were incubated with 2 µg/mL propidium iodide for 30 min as a marker of cell death. Slices were fixed and processed by immunohistochemistry for Aβ+ immunoreactivity using the Aβ antibody clone 6E10. (**A**) A representative image of Aβ-specific positive fluorescent staining in the ventral parts of the slices (green; AlexaFluor-488). (**B**) Propidium iodide stained the dead cells in the same field (red). (**C**) Slices were counterstained for the nuclear dye DAPI (blue). (**D**) A merged image was generated, which reveals that the staining for dead cells colocalizes with the Aβ+ immunoreactivity. A viable healthy cell can be visualized nearby, as indicated by the white arrow in all panels. Scale bar in D = 50 µm in (**A**–**D**).

**Figure 7 biomolecules-14-00165-f007:**
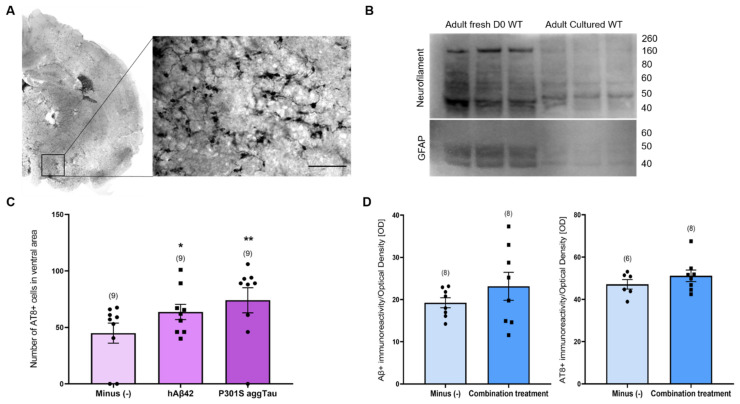
Culturing of adult slices from wild-type (WT) and transgenic (TG) animals. (**A**) Composite image of a half-brain adult slice shows AT8+ immunoreactivity after 9 weeks of culture with the magnified image showing representative AT8+ immunoreactivity in the ventral areas. (**B**) Acute fresh slices from 3 different adult WT animals (day 0) were collected and frozen immediately. Conversely, adult WT slices were cultured for 9 weeks and then compared via Western blotting. The acute slices display a strong signal for neurofilament and GFAP, whereas cultured slices show a severely reduced signal indicating the general decline of viability post-culture. (**C**) Adult slices from TG amyloid precursor protein _Swedish–Dutch–Iowa (APP_SDI) mice were incubated with collagen hydrogels containing human amyloid-beta 42 (hAβ42) or P301S aggregated tau (aggTau) for 9 weeks. A significantly increased number of AT8+ cells were quantified in the ventral areas between hAβ42- or P301S aggTau-loaded hydrogels and the empty hydrogels (minus group). The number of AT8+ cells were counted from the ventral regions and the values are reported as mean ± SEM. (**D**) Adult WT slices were supplemented with either scopolamine, wortmannin, MHY1485, lead and cadmium (50, 10, 50, 100, 100 nM final concentration, respectively) or normal media after 4 weeks of culture and then further cultured for 4 weeks. Slices were fixed and immunostained with the Aβ clone 6E10 or AT8 antibodies. No significant differences were found in the groups treated with the combination treatment and normal slice media. Images were quantified using ImageJ. Raw optical density measurements were inverted by correcting for slice background for each image such that 0 represents white and 255 represents black. Values are generated from an average of two images from left and right hemispheres per animal and values in parentheses indicate the number of analyzed animals. Statistical analyses were performed using a student’s *t*-test with equal variance, where *p* values < 0.05 represent significance versus the Minus (−) group (* *p* < 0.05, ** *p* < 0.01).

## Data Availability

The data that support the findings of this study are available on request from the corresponding author.

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
