# Peer review of "A Combination of Heavy Metals and Intracellular Pathway Modulators Induces Alzheimer Disease-like Pathologies in Organotypic Brain Slices"

_biomolecules, 2024, doi:10.3390/biom14020165_

Round 1

Reviewer 1 Report

Comments and Suggestions for Authors

This submission aims to present study on a-beta/tau related mechanisms using adult mice brain slices.  It supposed to describe methodology for reliable adult mice WT and Tg brain slice cultures, to study effects of exogenous a-beta and tau as well as aluminum and other xenobiotic metals/compounds on intracellular aggregates of these proteins.

Introduction should be shorter and separated subchapters interconnected going to main rationale.

Interesting point concerns very low concentrations of  pathogenic peptides as well as metals which appeared to be neuro-pathologically relevant.

Findings remain in agreement with  several previous studies performed on diverse cell culture, neonatal brain slices as well as in vivo models. Thereby they support claims that phenotypics of AD genes APP/tau mutations are aggravated  by metals or appropriate organic compounds. From this point submission looks somewhat confirmatory.

However, this pitfall is rescued by elegant experiments using adult brain slices that in fact are more relevant to neurodegeneration studies than fetal or newborn ones.

Therefore, I would see novelty of this paper  in this methodological aspect.

In the results section more accurate legends are required to make clear, whether adult or    neonatal slices are used.

Identification of intracellular accumulation of  A-beta as a cause of cell death in cultured slices  may be interesting finding.

Presented data allow no new conclusions on peptide-related mechanisms of AD. However, methodological aspect may be of interest.

Author Response

Ad referee 1

This submission aims to present study on a-beta/tau related mechanisms using adult mice brain slices.  It supposed to describe methodology for reliable adult mice WT and Tg brain slice cultures, to study effects of exogenous a-beta and tau as well as aluminum and other xenobiotic metals/compounds on intracellular aggregates of these proteins. 

Introduction should be shorter and separated subchapters interconnected going to main rationale.

Response: We agree. We have now revised the Introduction by reducing the text about Alzheimer’s disease (AD) and research methods to study AD and combining it into one paragraph. The text has also been rewritten more succinctly. We already have separate subchapters in the Introduction.

Interesting point concerns very low concentrations of pathogenic peptides as well as metals which appeared to be neuro-pathologically relevant. Findings remain in agreement with several previous studies performed on diverse cell culture, neonatal brain slices as well as in vivo models. Thereby they support claims that phenotypics of AD genes APP/tau mutations are aggravated by metals or appropriate organic compounds. From this point submission looks somewhat confirmatory. However, this pitfall is rescued by elegant experiments using adult brain slices that in fact are more relevant to neurodegeneration studies than fetal or newborn ones. Therefore, I would see novelty of this paper in this methodological aspect.

In the results section more accurate legends are required to make clear, whether adult or neonatal slices are used.

Response: Thank you for your comment. We would like to kindly point out that the legends of Figure 1, 3, 4 and 5 mention if we used adult or postnatal slices, including the age range. Figure 2 depicts post-mortem human brain sections and transgenic hTau expressing mouse sections. We have now added this information at the beginning of the figure legend to highlight it better.

Identification of intracellular accumulation of  A-beta as a cause of cell death in cultured slices  may be interesting finding. Presented data allow no new conclusions on peptide-related mechanisms of AD. However, methodological aspect may be of interest.

Reviewer 2 Report

Comments and Suggestions for Authors

The authors provide an interesting study which uses organotypic brain slices to valuate Amyloid beta and Tau pathologies upon various stimulators. The manuscript was well-written. But some points could be improved.

1. Figure 2, it will be great if the authors could also add images from WT mice as a control.

2.  The authors should provide some representative images for pharmacological manipulation instead of only showing a summary table 2.

3. Figure 3, how about postnatal TG mouse slices upon Aβ and Tau treatment.

Author Response

Ad referee 2

The authors provide an interesting study which uses organotypic brain slices to valuate Amyloid beta and Tau pathologies upon various stimulators. The manuscript was well-written. But some points could be improved.

  1. Figure 2, it will be great if the authors could also add images from WT mice as a control.

Response: Thank you for this comment. Figure 1 C-D has representative images of Aβ+ and tau+ immunoreactivity in ventral areas in slices from wild-type animals. The intention behind Figure 2 was to depict representative images from positive controls so we would prefer to keep the distinction between Figures 1 and 2 as it is. We hope that the reviewer is in agreement with the arrangement from our perspective.

  1. The authors should provide some representative images for pharmacological manipulation instead of only showing a summary table 2.

Response: Thank you for this comment. We have now added a supplementary figure, which includes representative images of each treatment with pharmacological agents. Please find this in the new Supplementary Figure 2.

  1. Figure 3, how about postnatal TG mouse slices upon Aβ and Tau treatment.

Response: We have already included data in Table 1 where the postnatal TG mouse slices were treated with a combination of Aβ and Tau in the collagen hydrogels. We only observed a slightly significant increase in the tau+ immunoreactivity in these slices compared to slices treated with only Aβ.

Additionally, a major aim of this study was to develop a model from wild-type mice as they can be bought and established in every lab worldwide. Our preliminary data with TG mouse slices did not point to a significant effect and response. Hence, we chose to focus on slices from wild-type mice.

Reviewer 3 Report

Comments and Suggestions for Authors

Dear Researchers

This is clearly a scientifically sound presentation of a relative unique research methodology.  There are some issues that l have that can be addressed without significant work as l believe that these experiments will have been done and can be presented as supplemental.

1)  As presented, in this extensive paper, there seems to be a lack of essential controls of specificity.  I do not accept that primary antibody delete is a rigorous control for IHC or western blots when using mouse monoclonal antibodies on mouse tissue.  This can cause all sorts of artefacts.  Need to use some mouse IgG as the control.  I find the images of so called aggregated tau throughout the neurons shown to be problematic without adequate controls.  The comparison with human is not convincing.

2)  In a paper such as this, l believe it is essential to biochemically characterize the starting materials.  What are the features of the purchased Ab and tau aggregates.  Running them on a non-denaturing gel or even SDS gel would be minimal requirement.  In addition, l think such western analyses at the end of the 8 weeks incubation is necessary.  I can not believe that there is not extensive degradation of these materials.  Prove me wrong.  In addition, characterizing start and end status of pTAU is needed.

3)  The presentation of the data tables when compared to the sample IHC images are hard to understand what are you actually counting.  It would be better for the readers to have a detailed panel supporting the numerical values being presented.  The table should be reworked to make the statistical differences more clear (highlight, scatter plot or something) to reader.

4).  The western blots shown are low quality.  Please annotate properly with molecular weight markers.  As mentioned above, these blots need to be probed with Ab antibodies and tau/ptau antibodies

5).  The overall readability of this paper is poor.  The introduction is more suitable for a review article.  It is possible to summarize the current models and their findings much more succinctly.  I accept that as you are going to be paying huge publishing costs, you can make it as long as you like but it does take away from the clearly sound work.  The same comment can be applied to the repetitive discussion.

6)  I would suggest including low magnification images of propidium iodide images at start and end of incubation period.  I find it hard to accept that there is not significant cell death occurring during the experimental period.  Please present.

7)  How many technical replicates did you include in these presented data

Author Response

Ad referee 3

This is clearly a scientifically sound presentation of a relative unique research methodology.  There are some issues that l have that can be addressed without significant work as l believe that these experiments will have been done and can be presented as supplemental.

  • As presented, in this extensive paper, there seems to be a lack of essential controls of specificity.  I do not accept that primary antibody delete is a rigorous control for IHC or western blots when using mouse monoclonal antibodies on mouse tissue.  This can cause all sorts of artefacts.  Need to use some mouse IgG as the control.  I find the images of so called aggregated tau throughout the neurons shown to be problematic without adequate controls.  The comparison with human is not convincing.

Response: Thanks for this comment. Multiple labs, including ours, have extensively used primary mouse antibodies used in this study in several publications and we trust these antibodies.

The primary antibodies used here (Aβ antibody clone 6E10, Tau5, AT8) have been characterized in Western blots in previous publications from our lab (Moelgg et al. 2021, Korde and Humpel, 2022). The blots clearly show that the antibodies can recognize our peptides/proteins, both individually and post-culture from slices.

We have utilized a mouse-on-mouse blocking solution (M.O.M., Vector Labs, MKB-2213-1) to specifically block endogenous mouse immunoglobulins in a mouse tissue sections. We believe that this extra blocking step prior to primary antibody incubation results in specific staining with the antibodies.

We are further convinced by our data as we performed several positive control experiments as shown in Figure 2 with post-mortem human brain tissue sections and hTau expressing mouse tissue sections. These data show that our antibodies of choice can successfully detect Aβ plaques and tau NFTs.

Based on your suggestion, we have included the requested controls in the new Supplementary Figure 1. We hope that this addresses your comment.

  • In a paper such as this, l believe it is essential to biochemically characterize the starting materials.  What are the features of the purchased Ab and tau aggregates.  Running them on a non-denaturing gel or even SDS gel would be minimal requirement.  In addition, l think such western analyses at the end of the 8 weeks incubation is necessary.  I can not believe that there is not extensive degradation of these materials.  Prove me wrong.  In addition, characterizing start and end status of pTAU is needed.

Response: Thank you for your comment. Yes we fully agree, but we have already extensively characterized the starting material in previous papers. We kindly refer the reviewer to carefully read our papers on Aβ (Moelgg et al., 2021) and on P301S aggregated tau (Korde and Humpel, 2022), where detailed characterizations are presented. These past publications are discussed in the manuscript.

Regarding the degradation of tau at the end of 9 weeks incubation, we have already characterized this in the Korde and Humpel paper. Unfortunately, we do not have data on phosphorylated tau degradation before and after 9 weeks incubation.

Based on your suggestion, we have included these data in the new Supplementary Figure 1. We hope that this addresses your comment.

  • The presentation of the data tables when compared to the sample IHC images are hard to understand what are you actually counting.  It would be better for the readers to have a detailed panel supporting the numerical values being presented.  The table should be reworked to make the statistical differences more clear (highlight, scatter plot or something) to reader.

Response: Thank you for this insight. We agree and the tables 1 and 2 have been converted to bar graphs with data points in a similar style to the bar graphs in Figure 7C-D. We hope that this style of data presentation is now clearer.

 4).  The western blots shown are low quality.  Please annotate properly with molecular weight markers.  As mentioned above, these blots need to be probed with Ab antibodies and tau/ptau antibodies

Response: We apologize for the low quality of the western blots. This is mainly arising from the imager and the condensation inside the equipment. Definitely, we have already added molecular markers for each blot. Based on your suggestion, we have included additional control data in the new Supplementary Figure 1. We hope that this addresses your comment.

5).  The overall readability of this paper is poor.  The introduction is more suitable for a review article.  It is possible to summarize the current models and their findings much more succinctly.  I accept that as you are going to be paying huge publishing costs, you can make it as long as you like but it does take away from the clearly sound work.  The same comment can be applied to the repetitive discussion.

Response: We are sorry that the reviewer thinks that the readability is poor. This is unexpected to us, as both reviewers 1 and 2 did not criticize this, but only asked for a reduced Introduction section. We have reduced the text about Alzheimer’s disease and research methods to study AD and combined it into one paragraph. We have also summarized the current models more succinctly. The discussion is shortened and rewritten. We hope that this aids to the readability of the manuscript.

6)  I would suggest including low magnification images of propidium iodide images at start and end of incubation period.  I find it hard to accept that there is not significant cell death occurring during the experimental period.  Please present.

Response: Thank you for your suggestion. PI is a very rough method and gives an estimate on cell death. There is no need to show PI from start to beginning, as we only wanted to compare the end point with the different treatments. It is fully clear that slices partly undergo cell death from start of culturing, especially at the edges. In this study, PI is used as a strict end-point only in co-localization experiments. For a measure of cell death/viability, we compare the samples from fresh uncultured adult slices to cultured adult slices. Based on your suggestion, we have included additional controls in the new Supplementary Figure 1. We hope that this addresses your comment.

7)  How many technical replicates did you include in these presented data?

Response: In each well, we collected two slices per treatment. From these two slices, we extracted data from both left and right ventral regions. The values from both hemispheres were averaged per slice and subsequently, the values from both slices were averaged to get to a final result per animal.

For the data on spreading in postnatal wild-type and transgenic slices, we extracted values from 6 different animals/replicates.

For the data in postnatal wild-type   slices   with   pharmacological   manipulation,   we   extracted   values   from   4-7 animals/replicates. Some slices were excluded due to poor appearance of the slices after the culturing period. This is why some groups have data from 4 analyzed animals. However, for the combination treatments, the data comes from 6 different animals. 

For the data from adult slices, we extracted values from 6-9 different animals/replicates.

We have now explained this better in the Methods under the Data analysis and statistics section.

Round 2

Reviewer 3 Report

Comments and Suggestions for Authors

As mentioned, this is an interesting study but some of the fundamentals should be validated.

In methods (2.3 section) you do not describe or reference how the tau and abeta are prepared for your experiments.  I do notice that Abcam do not describe how their tau becomes aggregated.  This is an important detail for your experiments.  The company describes increased Thio T reactivity when this notoriously insoluble material is mixed with monomeric.  Is this the mixture you used.  In light of the interpretation of your data, this should be clarified to establish relevance of findings to biological systems.

Similarly, your supplemental WB of your Abeta peptide confuses the whole findings.  Since when are Abeta monomers  40 kD.  In your slice WB for Abeta, you do not show the low molecular weight forms of Abeta as the gel was not correctly run or a non-specific band.  Since when is Abeta 160 kD.  Although 6e10 is supposed not to react with mouse APP, it can react weakly with such bands depending on concentration of antibody used in experiments.  Did you add monomeric Abeta to your cultures or did you aggregate it before hand.  I see no mention of this.

You should provide a clear version of your revised text along with the marked up when resubmitting.   Reading a scored version is not a pleasant experience.

Author Response

Response to referee 3:

We thank this referee for the helpful comments and apologize that we provided an inaccurate Western Blot for beta-Amyloid in the Supplementary Figure 1.

As mentioned, this is an interesting study but some of the fundamentals should be validated.

In methods (2.3 section) you do not describe or reference how the tau and abeta are prepared for your experiments.  I do notice that Abcam do not describe how their tau becomes aggregated.  This is an important detail for your experiments.  The company describes increased Thio T reactivity when this notoriously insoluble material is mixed with monomeric.  Is this the mixture you used.  In light of the interpretation of your data, this should be clarified to establish relevance of findings to biological systems.

Response: We use the P301S aggregated tau directly as we purchase it from Abcam. This is how we used it in our previous publication (Korde and Humpel, 2022) and since this present study builds on that work, we chose to use the same experimental approach. Similarly, we used the human Aβ42 as purchased from Innovagen and based the work on a previous publication (Moelgg et al., 2021).

For the aggregation of amyloid beta and tau, we used this data only in Supplementary Figure 1 where full-length tau and human Ab42 were aggregated. We have now added details of the procedure to the Methods section under Peptides/proteins.

Similarly, your supplemental WB of your Abeta peptide confuses the whole findings.  Since when are Abeta monomers  40 kD.  In your slice WB for Abeta, you do not show the low molecular weight forms of Abeta as the gel was not correctly run or a non-specific band.  Since when is Abeta 160 kD.  Although 6e10 is supposed not to react with mouse APP, it can react weakly with such bands depending on concentration of antibody used in experiments.  Did you add monomeric Abeta to your cultures or did you aggregate it before hand.  I see no mention of this.

Response: Yes, you are correct. Unfortunately, this was a labelling error and it is a pity that we did not recognize that. Sure, beta-amyloid monomer has a size of four kDa. The rectified Supplementary Figure 1 has been uploaded. We sincerely apologize for this error and inconvenience and hope that the reviewer accepts the changes.

We used monomeric human Aβ42 as a load inside collagen hydrogels and the concentration of 6e10 used for Western blot detection was 1:5000. You are right; it could be murine APP, which has been added to the figure legend of Supplementary Figure 1.

You should provide a clear version of your revised text along with the marked up when resubmitting.   Reading a scored version is not a pleasant experience.

Response: We apologize, but the system only allowed uploading one single Word document, however, we previously uploaded also a cleared PDF. This new version is easier to read. Thank you!